# A Diffusive Data Augmentation Framework for Reconstruction of Complex Network Evolutionary History

## Abstract

The evolutionary processes of complex systems contain critical information about their functional characteristics. The generation time of edges can reveal the historical evolution of various networked complex systems, such as protein-protein interaction networks, ecosystems, and social networks. Recovering these evolutionary processes holds significant scientific value, such as aiding in the interpretation of the evolution of protein-protein interaction networks. However, the scarcity of temporally labeled network data poses challenges for predicting edge generation times under current network structures, leading to issues of insufficient data and significant differences between training and prediction networks. To address this, we introduce a diffusion model that learns the generative mechanisms of networks, producing sufficient augmented network data to effectively mitigate issues of limited and incomplete data. Experimental results demonstrate a 13.7% improvement in prediction accuracy using our approach. Moreover, the model can uniformly predict edge generation times across different types of networks, eliminating the need to retrain the model for each specific network, thus significantly enhancing generalization capability and efficiency.

## 1 Introduction

The core challenge in reconstructing the evolution of complex networks is inferring the order in which edges are formed in dynamic graphs, thus reproducing the network's evolutionary process (Boccaletti et al. (2006); Liao et al. (2017)). This task is intrinsically tied to the dependency between network structure and edge formation sequence, as accurately inferring this order can reveal the underlying dynamic mechanisms of the network, as illustrated in Figure 1. Successfully reconstructing network evolution has significant applications across various domains, such as social network analysis, traffic system optimization, and biological network research (Barabâsi et al. (2002); Wang et al. (2023); Seferbekova et al. (2023)). These fields require a deep understanding of how networks evolve over time to predict future trends and implement effective interventions or optimizations.

However, reconstructing complex network evolution is a typical small-sample problem (Margeloiu et al. (2023)). The real-world mechanisms driving network evolution are often highly complex and unclear, making theoretical reconstruction exceedingly difficult. Machine learning-based algorithms offer promising solutions to this problem (Banerjee et al. (2023)). For instance, methods like those in Nature Communications 2024 proposed by Wang et al. (2024) employ graph embedding and machine learning to automatically learn the spatiotemporal structures of networks. Nonetheless, these algorithms face significant challenges due to data scarcity. The dynamic evolution of real networks is rarely fully observable, making it difficult to acquire high-quality labeled data. As a result, models typically rely on a limited number of samples for training, which not only leads to overfitting but also severely hampers their generalization and transferability, resulting in suboptimal performance across different networks and scenarios.

To address these challenges, we propose a graph generation-based sample augmentation method aimed at enhancing the reconstruction of network evolution. By generating dynamic graph samples that adhere to the actual evolutionary patterns of real networks, this method supplements the original small training dataset, significantly improving the model's reconstruction capabilities. However,

Figure 1: Evolution diagram of a complex network illustrating the dynamic process of node and edge formation over time.

this approach introduces several technical challenges. First, we must jointly model the distribution of both network structure and edge formation times. Existing graph generation methods primarily focus on capturing the temporal-structural associations within a specific network, rather than learning a generalized joint distribution of structure and formation time. While these methods can generate temporal networks within a fixed network type and size, they lack the ability to generalize across different network types and scales. To ensure the augmentation method is applicable to various networks, we need to generate flexible samples for networks of different sizes, domains, and mechanisms.

Our proposed solution is a sample augmentation framework based on dynamic graph generation using diffusion models. By leveraging diffusion models, we effectively capture the intricate dependencies between network structure and edge formation times, achieving accurate modeling of their joint distribution. Moreover, diffusion models possess the capacity to generate diverse samples, enabling them to produce augmented samples that follow the evolutionary patterns of various complex networks. This enhances both the model's generalization ability and its performance in network reconstruction.

Ultimately, our approach demonstrates substantial improvements across a range of scenarios, regardless of the size, mechanism, or domain differences between the target and source networks. The results indicate that sample augmentation plays a crucial role in complex network evolution reconstruction, particularly when addressing small-sample and data-scarce conditions, providing a powerful tool to improve model performance.

## 2 PRELIMINARIES

In the study of complex networks, the generation time of edges is a critical factor influencing network evolution. Our prediction task aims to learn the relationship between network structure and edge generation time, given a temporal network. Specifically, we focus on how to infer the generation order of edges in the absence of time labels.

Our ultimate goal is to predict the generation time of each edge for a pure network structure. To simplify the task, we specify it as follows: for any two edges $e_i$ and $e_j$ in the network, we need to predict their generation order, determining which edge is generated first and which is generated later. By inferring the order of all edges, we can construct the overall generation sequence of edges in the network, thereby gaining a comprehensive understanding of the network's evolutionary process.

### 2.1 MATHEMATICAL DEFINITION

1. **Temporal Network Representation**: Let the temporal network $G$ consist of a set of nodes $V$ and a set of edges $E$. Each edge $e_k \in E$ represents a connection between a pair of nodes $(u_k, v_k)$ and has an associated generation time $t_k$.

2. **Edge Generation Order**: For any two edges $e_i = (u_i, v_i)$ and $e_j = (u_j, v_j)$, we define the generation order relation $R(e_i, e_j)$ as follows:

$$R(e_i, e_j) = \begin{cases} 1, & \text{if } t_i < t_j \\ 0, & \text{if } t_i \geq t_j \end{cases}$$

Here, $R(e_i, e_j) = 1$ indicates that edge $e_i$ is generated before edge $e_j$, while $R(e_i, e_j) = 0$ indicates that the generation time of edge $e_i$ is later than or equal to that of edge $e_j$.

3. **Edge Generation Time Prediction**: Our task can be expressed as a binary relation prediction problem, where the objective is to predict the edge generation order $R(e_i, e_j)$ by learning the structural features of the network $S(G)$:

$$\hat{R}(e_i, e_j) = f(S(G), e_i, e_j)$$

In this context, $\hat{R}(e_i, e_j)$ is the predicted generation order, and the function $f$ represents the model used to infer the generation order based on network structural features.

4. **Construction of Edge Generation Sequence**: Once the generation order for all edges is predicted, we can obtain the overall edge generation sequence $T$, expressed as:

$$T = \{e_{k_1}, e_{k_2}, \ldots, e_{k_m}\}$$

Here, $k_1, k_2, \ldots, k_m$ are the indices of edges arranged according to the predicted generation order.

Through the above description and mathematical definitions, our task clearly demonstrates how to infer edge generation times based solely on structural information in the absence of time labels, thus providing robust support for the evolutionary prediction of complex networks.

## 3 METHOD

### 3.1 MODEL FRAMEWORK FOR EDGE TIME PREDICTION

The framework for predicting the order of two edges in a network is structured as follows. Initially, we utilize an embedding representation learning method to obtain a representation vector for each edge in the network. Let us denote the edges as $e_1$ and $e_2$, with their respective embedding vectors represented as $\mathbf{h}_1$ and $\mathbf{h}_2$.

Each embedding vector is subsequently fed into a fully connected neural network comprising three layers. The architecture is defined as follows:

Input Layer: The first layer receives the input vectors $\mathbf{h}_1$ and $\mathbf{h}_2$, which are concatenated to form a single input vector $\mathbf{h} = [\mathbf{h}_1; \mathbf{h}_2] \in \mathbb{R}^d$, where $d$ is the dimension of the embedding vectors.

Hidden Layer: The second layer is a hidden layer with a dimensionality of $\frac{2}{3}d$. This layer applies a linear transformation followed by a ReLU activation function, defined as:

$$\mathbf{h}_{\text{hidden}} = \text{ReLU}(\mathbf{W}_1 \mathbf{h} + \mathbf{b}_1),$$

where $\mathbf{W}_1 \in \mathbb{R}^{\frac{2}{3}d \times d}$ and $\mathbf{b}_1 \in \mathbb{R}^{\frac{2}{3}d}$ are the weight matrix and bias vector for the hidden layer, respectively.

Output Layer: The output layer consists of two neurons that produce a scalar output $z_1$ and $z_2$:

$$\begin{bmatrix} z_1 \\ z_2 \end{bmatrix} = \mathbf{W}_2 \mathbf{h}_{\text{hidden}} + \mathbf{b}_2,$$

where $\mathbf{W}_2 \in \mathbb{R}^{2 \times \frac{2}{3}d}$ and $\mathbf{b}_2 \in \mathbb{R}^2$ are the weights and biases for the output layer.

After obtaining the outputs $z_1$ and $z_2$, we apply the softmax function to convert these into a probability distribution representing the likelihood that each edge was generated first:

$$p(e_1 \text{ before } e_2) = \frac{e^{z_1}}{e^{z_1} + e^{z_2}},$$

$$p(e_2 \text{ before } e_1) = \frac{e^{z_2}}{e^{z_1} + e^{z_2}}.$$

These probabilities reflect the generation times of the edges, normalized to ensure that they sum to one.

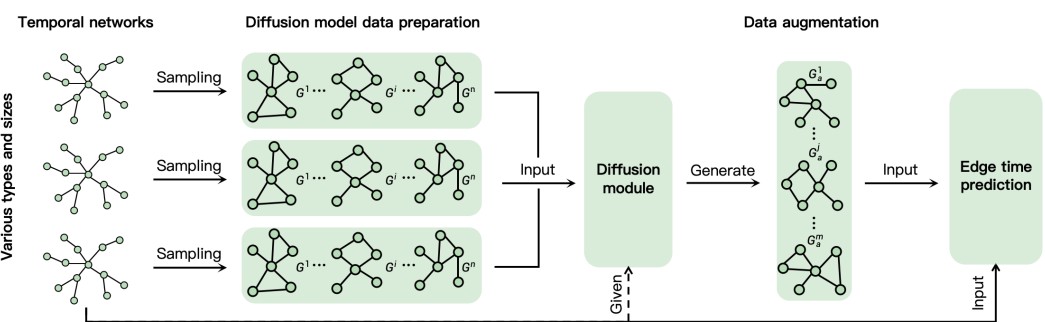

Figure 2: Schematic framework of temporal network augmentation using diffusion models.

The loss function $L$ is defined as the categorical cross-entropy loss between the predicted probabilities and the true labels $y \in \{0, 1\}$, where $y = 1$ indicates that $e_1$ occurs before $e_2$:

$$L = -y \log(p(e_1 \text{ before } e_2)) - (1 - y) \log(p(e_2 \text{ before } e_1)).$$

Additionally, we incorporate $L2$ regularization to mitigate overfitting. The regularization term is defined as:

$$R = \lambda \left( \|\mathbf{W}_1\|^2 + \|\mathbf{W}_2\|^2 \right),$$

where $\lambda$ is the regularization strength. Thus, the final loss function to be minimized is:

$$\mathcal{L} = L + R.$$

## 3.2 Data Augmentation for Edge Time Prediction Model

As shown in Figure 2, we sample networks with temporal labels, varying in type and size, to train the diffusion model. The diffusion model generates networks resembling the target temporal networks, augmenting the original training data and improving edge generation time prediction accuracy. Two distinct augmentation frameworks address the limitations of existing methods.

1. Mitigating Transfer Learning Challenges: Existing models often struggle with transfer learning due to structural and temporal discrepancies between training and target networks. To resolve this, we employ the diffusion model to generate temporal networks that closely resemble the target network. The model is trained exclusively on these generated networks, ensuring alignment between training and target data, which eliminates transfer learning issues and enhances predictive performance.

2. Enhancing Robustness and Fitting Ability: A single network often provides insufficient training data. When multiple networks are combined, structural inconsistencies can degrade performance. To overcome this, we generate additional temporal networks similar to the training network using the diffusion model. By combining the original and generated networks, we enrich the training set, improving the model's robustness and fitting capability across similar network structures.

## 3.3 Construction of Training Dataset for the Diffusion Model

The construction of the training dataset for our diffusion model is a crucial step in enhancing the edge order prediction task. Our approach begins with a labeled temporal network, which serves as the foundational dataset. The methodology can be outlined as follows:

Sampling Strategy: We employ a systematic sampling technique to generate multiple training network samples from the original labeled temporal network. Specifically, we start by sampling a subset of edges, beginning with 50% of the total edges and progressively increasing to include all edges in the network. This approach allows us to explore various network densities and their impacts on the learning process.

Creation of Diverse Samples: For each sampling iteration, we construct a new training network by retaining the sampled edges and discarding the remaining edges. This results in 100 distinct training

network samples, each representing a unique configuration of the original temporal network. As shown in Figure 2, $n = 100$, and we sampled $G^1, \ldots, G^{100}$ networks. By varying the percentage of edges retained, we ensure a diverse representation of network structures, which is essential for robust model training.

### 3.4 Graph Denoising Diffusion Model: TopoEvoDiff

Based on the training data sampled from the real temporal network, we utilize the graph diffusion models to learn the distribution of the topology evolutions conditioned on the final topology structure. We refer to our model as the *Diffusion Model for Generating Topology Evolution History*, abbreviated as **TopoEvoDiff**. The graph diffusion models are adapted to ordered edges diffusion based on the final topology structure, where the structural features are encoded into the node features, such as node embeddings learned through structural learning.

The graph diffusion model consists of two main procedures: the forward diffusion process $q$ and the reverse denoising process $p$. In the forward diffusion process, the network orders are disrupted by Gaussian noise step by step and reach to pure noise. The forward diffusion process is utilized to construct the training data for the denoising networks used in the reverse denoising process. The computation is defined as: $F$ should mean the weights and $ij$ should mean the indices of the edges

$$q(F_{ij}^t | F_{ij}^{t-1}) = \mathcal{N}(F_{ij}^t; \sqrt{1 - \beta_t} F_{ij}^{t-1}, \beta_t \mathbf{I}),$$

$$q(F_{ij}^1, ..., F_{ij}^T | F_{ij}^0) = \prod_{t=1}^{T} q(F_{ij}^t | q^{t-1}). \tag{1}$$

where introduce the meaning of symbols, t: diffusion steps, N: normal distribution, betat: noise level at step t, I: identity matrix. The reverse denoising process is to recover the origin edge orders from the pure noise, with the guidance of the final topology structure. In each step, the noised to be removed from the noisy data is predicted by the denoising networks. The denoising networks used in the reverse process are graph transformers (Dwivedi & Bresson, 2020). In the denoising process, node features serve as guidance to direct the data generation, remaining unchanged throughout. In each step of the denoising process, the noisy edges and the node features used as guidance are fed into the graph transformer as the node inputs and edge inputs, and the small noise that needs to be removed from the noisy edges in a single step is output. The computation of the reverse denoising process is defined as:

$$p_\theta(\mathbf{F}^{t-1} | \mathbf{F}^t, \mathcal{C}_\mathcal{R}) = \mathcal{N}(\mathbf{F}^{t-1}; \mu_\theta(\mathbf{F}^t, t, \mathcal{C}_\mathcal{R}), (1 - \bar{\alpha}^t)\mathbf{I}), \tag{2}$$

where

$$\mu_\theta(\mathbf{F}^t, t, \mathcal{C}_\mathcal{R}) = \frac{1}{\sqrt{\alpha_t}}\left(\mathbf{F}^t - \frac{\beta_t}{\sqrt{1 - \bar{\alpha}_t}}\epsilon_\theta(\mathbf{F}^t, t, \mathcal{C}_\mathcal{R})\right), \tag{3}$$

The denoising networks are trained to minimize the reconstruction loss between the predicted noise and the true noise. The loss function we adopted is the same as denoising diffusion probabilistic models (DDPMs) (Ho et al., 2020), which is the mean squared error between the predicted noise and the true noise. The loss function is defined as:

$$\mathcal{L} = \mathbb{E}_{t, \epsilon \sim \mathcal{N}(0, \mathbf{I})}\left[\|\epsilon - \epsilon_\theta(\mathbf{F}^t, t, \mathcal{C}_\mathcal{R})\|_2^2\right] \tag{4}$$

where $\|\dot{\|}\|$ denotes the $L2$ norm.

When the model is trained, we can sample the temporal networks with similar distribution as the original dataset by the graph diffusion model. First, a pure Gaussian noise is sampled and then the denoising networks iteratively predict the noise to be removed, and the ordered edges will be obtained from the weights of the sampled network gradually. The generated temporal networks are used to augment the original dataset for the edge order prediction model.

## 4 Experiments

### 4.1 Datasets

This study utilizes three synthetic models for simulating networks: the Barabási–Albert (BA) model, the Popularity-similarity-optimization (PSO) model, and the Fitness model. Each model effectively

simulates real-world network evolution. Specifically, we generate BA100 and Fitness100 networks, each with 100 nodes, while the PSO model yields PSO100 with 100 nodes and PSO200 with 200 nodes.

- **Barabási–Albert (BA) model**: A *preferential attachment-based growth model* that captures the *scale-free* property of networks.

- **Fitness model**: A variant of the BA model, incorporating node *fitness* to influence connection probabilities.

- **Popularity-similarity-optimization (PSO) model**: Combines *popularity* and *similarity* for establishing connections, as detailed in Appendix A.

We employ four real-world datasets. In addition to synthetic datasets, we use two protein-protein interaction (PPI) networks, as well as two collaboration networks derived from the American Physical Society (APS) publication data.

- **Protein-Protein Interaction (PPI) Networks**: The *Fruit Fly (Drosophila melanogaster)* and *Worm (Caenorhabditis elegans)* PPI networks model the interactions between proteins in these organisms. In both datasets, nodes represent proteins, and edges signify interactions between them. The temporal aspect of these networks provides valuable biological insights, making them essential for testing edge generation time prediction and temporal network reconstruction methods.

- **Collaboration Networks**: The *Complex Network* (PACS = 89.75, 2001-2010) and *Thermodynamics* (PACS = 05.70, 1978-2010) datasets represent academic collaborations. In both datasets, nodes correspond to authors, and edges are formed when two authors co-author at least two papers. The edge generation time is determined by the date of the earliest co-authorship between two authors. These datasets offer complex collaboration patterns critical for evaluating temporal network analysis.

## 4.2 BASELINE MODELS

In this work, we compare our proposed diffusion-based data augmentation method for temporal network generation with several state-of-the-art baseline models. These baselines utilize different embedding methods and data augmentation strategies, providing a comprehensive comparison across various approaches. Below are the baseline models used in our experiments:

- **CPNN + Node2Vec**: This baseline combines the comparative paradigm neural network (CPNN) approach from Nature Communications Wang et al. (2024) with Node2Vec Grover & Leskovec (2016), which optimizes random walks to capture both homophily and structural equivalence in the network. It aims to predict edge generation order based on these learned embeddings.

- **CPNN + DeepWalk**: This baseline follows the CPNN framework but uses DeepWalk as the embedding method Perozzi et al. (2014). DeepWalk generates node embeddings via random walks, focusing on capturing node proximity.

- **CPNN + LINE**: In this setup, the CPNN method is paired with LINE (Large-scale Information Network Embedding) Tang et al. (2015), which preserves both first- and second-order proximities of nodes, capturing local and global network structures for edge prediction.

- **TIGGER + Node2Vec**: This baseline is based on TIGGER Gupta et al. (2022), which generates synthetic temporal networks for data augmentation. Node2Vec is used for embedding, allowing us to compare different augmentation techniques.

- **TIGGER + DeepWalk**: The final baseline combines TIGGER with DeepWalk, allowing us to evaluate the performance of DeepWalk in temporal augmentation scenarios.

- **TIGGER + LINE**: The TIGGER method is paired with LINE in this baseline. LINE's ability to capture local and global structures is tested to assess its performance in temporal network augmentation.

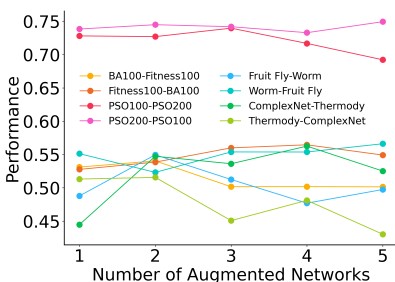

Figure 3: Impact of the number of augmented networks on predictive performance.

### 4.3 Effectiveness of Data Augmentation

In this experiment, conducted under the CPNN framework, we generated augmented temporal networks using both the TIGGER method and our diffusion-based approach. These generated samples were combined with the original training data to enhance the model. The results in Table 1 demonstrate that our diffusion-based data augmentation method significantly improves the prediction accuracy for edge generation order across various networks. For example, in the BA100 to Fitness100 task, our approach outperforms both CPNN and TIGGER baselines, achieving up to a 5.6% improvement with the LINE embedding. Similarly, when applied to more complex networks like PSO100 and PSO200, our method shows a substantial increase of 12.9%. Overall, our method consistently boosts performance across different embeddings, with average improvements ranging from 3% to 12%, confirming its effectiveness in enhancing temporal network learning.

Table 1: Performance of Different Methods for Predicting Between Networks

| TrainingNet | TargetNet | Node2Vec | | | DeepWalk | | | LINE | | |
|---|---|---|---|---|---|---|---|---|---|---|
| | | CPNN | TIGGER | TopoEvoDiff | CPNN | TIGGER | TopoEvoDiff | CPNN | TIGGER | TopoEvoDiff |
| BA100 | Fitness100 | 0.5097 | 0.5358 (5.1%) | 0.5312 (4.2%) | 0.5267 | 0.5384 (2.2%) | 0.5426 (3.0%) | 0.5189 | 0.5368 (3.4%) | 0.5478 (5.6%) |
| Fitness100 | BA100 | 0.4944 | 0.5234 (5.9%) | 0.5277 (6.7%) | 0.5144 | 0.5195 (1.0%) | 0.5274 (2.5%) | 0.5148 | 0.5254 (2.1%) | 0.5448 (5.8%) |
| PSO100 | PSO200 | 0.7183 | 0.7208 (0.3%) | 0.7281 (1.4%) | 0.7268 | 0.7249 (-0.3%) | 0.7589 (4.4%) | 0.6103 | 0.6267 (2.7%) | 0.6447 (5.6%) |
| PSO200 | PSO100 | 0.7136 | 0.7294 (2.2%) | 0.7384 (3.5%) | 0.7488 | 0.7808 (4.3%) | 0.7697 (2.8%) | 0.5562 | 0.5778 (3.9%) | 0.6281 (12.9%) |
| Fruit Fly | Worm | 0.4613 | 0.4658 (1.0%) | 0.4879 (5.8%) | 0.4470 | 0.4611 (3.2%) | 0.4723 (5.7%) | 0.4868 | 0.5144 (5.7%) | 0.5250 (7.8%) |
| Worm | Fruit Fly | 0.5242 | 0.5360 (2.3%) | 0.5513 (5.2%) | 0.5157 | 0.5598 (8.5%) | 0.5420 (5.1%) | 0.5016 | 0.4873 (-2.9%) | 0.5315 (6.0%) |
| ComplexNet | Thermody | 0.5045 | 0.5190 (2.9%) | 0.5148 (2.0%) | 0.4880 | 0.4823 (-1.2%) | 0.5046 (3.4%) | 0.5023 | 0.5097 (1.5%) | 0.5176 (3.0%) |
| Thermody | ComplexNet | 0.4188 | 0.4815 (15.0%) | 0.5132 (22.6%) | 0.5185 | 0.4931 (-4.9%) | 0.5412 (4.4%) | 0.4901 | 0.5099 (4.1%) | 0.5321 (8.6%) |

### 4.4 Selecting the Number of Augmented Samples

In this section, we evaluate the impact of generating multiple augmented networks on the performance of our edge generation time prediction model. Specifically, we investigate the effect of varying the number of augmented networks, ranging from 1 to 5. By analyzing how the number of augmented networks influences the model's predictive accuracy, we aim to understand the relationship between augmentation quantity and model performance.

As shown in Figure 3, there is no clear trend that increasing the number of augmented networks significantly enhances performance. A possible explanation is that too many augmented samples may dilute the weight of the original training data, thereby reducing the overall effectiveness of augmentation. Additionally, considering that a higher number of samples also increases the training time, augmenting with one or two additional networks seems to provide a reasonable balance between performance improvement and training efficiency.

### 4.5 Joint Training and Target Network Generation

In this experiment, our setup differs from previous experiments. In the earlier studies, training data consisted solely of 100 samples drawn from the original temporal network. In contrast, this experiment utilizes multiple types of networks to sample a diverse array of training data across different types and sizes, enabling the training of the diffusion model.

In this context, **Target** refers to an augmented network similar to the target network, generated using the diffusion model, which is used to predict edge generation times for the target network. By training the model on the generated temporal networks, we can effectively predict the target network. **Joint**, on the other hand, represents the augmented samples created by the diffusion model using training temporal networks. This approach combines the original training network with the augmented samples for model training before making predictions on the target network. Unlike the model discussed in Section 4.3, this chapter's model leverages cross-type and cross-size network training, aimed at enhancing the model's generalization capabilities.

The specific results of the experiments are illustrated in Table 2. We observe that under Node2Vec, the Joint approach achieves a significant improvement, averaging (33.05% + 10.28% + 3.89% + 6.7%) / 4, which is approximately 13.73%. This highlights the effectiveness of our augmentation strategy in enhancing model performance.

Table 2: Performance of Different Network Generation Methods with Improvement Ratios

| TrainingNet | TargetNet | Node2Vec | | DeepWalk | | LINE | |
|---|---|---|---|---|---|---|---|
| | | Target | Joint | Target | Joint | Target | Joint |
| BA100 | Fitness100 | 0.6613 (29.76%) | 0.6782 (33.05%) | 0.5913 (12.17%) | 0.5574 (5.82%) | 0.5686 (9.64%) | 0.5858 (13.00%) |
| Fitness100 | BA100 | 0.5256 (6.31%) | 0.5453 (10.28%) | 0.5406 (5.08%) | 0.5606 (8.96%) | 0.548 (6.67%) | 0.5686 (10.63%) |
| PSO100 | PSO200 | 0.7319 (1.90%) | 0.7463 (3.89%) | 0.7463 (2.63%) | 0.7519 (3.53%) | 0.6204(1.65%) | 0.6619 (8.45%) |
| PSO200 | PSO100 | 0.7374 (3.33%) | 0.7614 (6.70%) | 0.7931 (5.93%) | 0.7906 (5.57%) | 0.6313 (12.86%) | 0.5694 (2.40%) |

## 4.6 SIMILARITY BETWEEN AUGMENTED NETWORKS AND ORIGINAL NETWORKS

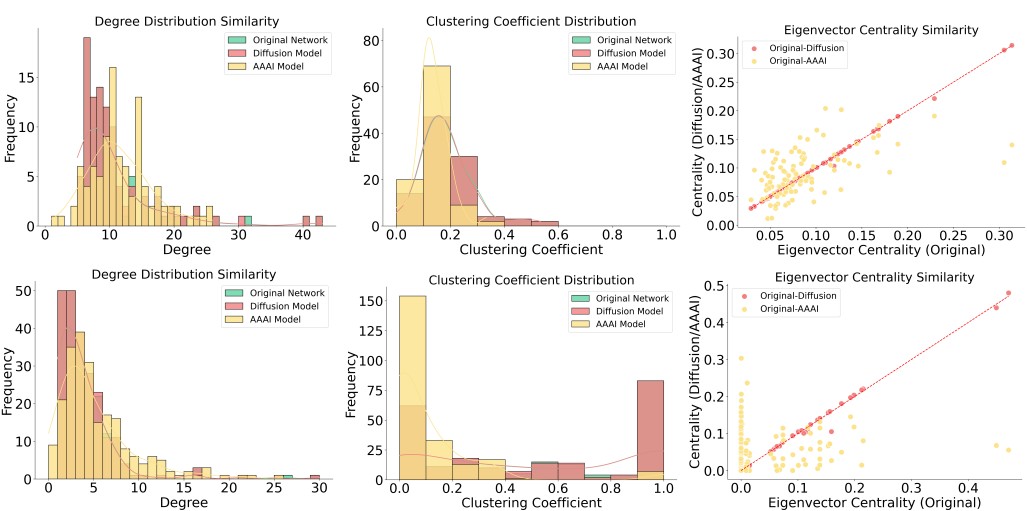

Figure 4: Similarity between the generated networks and the original networks.

This section evaluates the structural and dynamic similarity between augmented networks and the original networks. We employ various metrics, including degree distribution, eigenvector centrality, and clustering coefficient distribution, to assess similarity. The analysis reveals that the degree distributions of the networks generated by TopoEvoDiff closely align with those of the original networks. This indicates a strong preservation of structural characteristics. Furthermore, the eigenvector centrality measures show a robust correlation between nodes' importance in both networks, confirming the retention of critical structural roles. Lastly, clustering coefficient distributions closely match between the generated and original networks, reflecting consistent local clustering tendencies.

Figure 4 illustrates these results, with the top three graphs depicting findings from the BA100 network and the bottom three representing the real-world Complex Network. The results clearly demonstrate that networks generated by TopoEvoDiff exhibit a higher degree of similarity to the original networks compared to those produced by the TIGGER method.

## 5 RELATED WORK

### 5.1 TEMPORAL NETWORKS IN COMPLEX NETWORKS

Recent advancements in temporal networks have significantly enriched complex network analysis. Early models such as Erdős-Rényi and Barabási-Albert (Karoński & Ruciński (1997); Watts & Strogatz (1998)) mainly addressed static network properties, offering limited applicability to dynamic systems. Holme & Saramäki (2012) provided foundational concepts like temporal paths, establishing a basis for subsequent research. Karsai et al. (2012) highlighted bursty interaction patterns in real-world networks, emphasizing temporal heterogeneity. Masuda et al. (2017) extended random walk theory to temporal networks, crucial for understanding diffusion processes. Mucha et al. (2010) introduced methods for detecting communities in dynamic structures, while Millidge et al. (2024) integrated predictive models for forecasting interactions. Rossetti & Cazabet (2018) reviewed evolving community detection approaches, showcasing the need to address both temporal and topological dynamics in analysis.

### 5.2 DYNAMIC GRAPHS AND TEMPORAL NETWORK GENERATION

Recent developments in temporal network generation have significantly advanced the field, focusing on capturing both the structural and temporal dynamics of evolving networks. Zhou et al. (2020) improved temporal modeling by transforming temporal interactions into static graphs, while Zeno et al. (2021) utilized motif-based approaches for generating time-dependent structures, though it assumed fixed rates and lacked adaptability to real-world temporal fluctuations. Gupta et al. (2022) introduced TIGGER leveraging temporal point processes, combining both inductive and transductive learning to handle large-scale, complex networks with superior efficiency and accuracy. This progression highlights the growing ability of temporal models to replicate the nuanced evolution of real-world dynamic systems.

### 5.3 DATA AUGMENTATION VIA GRAPH DIFFUSION MODELS

Employing generative methods such as generative adversarial networks (GANs) to learn a distribution based on a small sample of data and subsequently conduct sampling to obtain a large quantity of training samples is a commonly adopted method (Zhang et al., 2022; Abdollahzadeh et al., 2023; Zhang et al., 2018; Fu et al., 2024). Diffusion models have recently shown significant advantages in performance, and have done well on graph generation tasks (Vignac et al., 2022; Niu et al., 2020; Jo et al., 2022), making them suitable for solving the problem of small-sample learning in reconstructing network evolutions. Specifically, Niu et al. (2020) first propose to introduce the score-based diffusion model to solve the graph generation problem. The Gaussian noise is adopted to construct the diffusion process, leading to poor sparsity generated graphs. Jo et al. (2022) explore jointly modeling the nodes and edges with semantic features. Vignac et al. (2022) use multinomial noise to make the diffusion process operating in discrete space, which greatly improves the performance. Therefore, we can see that using graph diffusion models to generate sufficient samples to enhance edge order prediction performance is promising.

## 6 CONCLUSION

Our work focuses on predicting edge generation times from given network structures, which aids in understanding network evolution and forecasting future developmental trends. We leverage diffusion models to generate additional temporal networks, addressing challenges related to limited sample sizes that often lead to overfitting and poor generalization. By augmenting the training data, we significantly enhance the model's fitting capability. Furthermore, by generating augmented data for the target network, we mitigate transferability issues, resulting in an improvement of 13%. Importantly, our approach allows for the training of a unified model capable of generating augmented data across all network types, maximizing flexibility and applicability in diverse contexts.

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

## A  SYNTHETIC DATASETS

In the supplementary materials, we provide further details on the three network models: BA model, PSO model, and Fitness model.

- **Barabási–Albert (BA) model**: The BA model generates networks through *preferential attachment*. Starting with a small fully connected graph, each new node added to the network connects to an existing node with probability $p \propto k$, where $k$ is the degree (number of connections) of the existing node. This model reflects the *power-law degree distribution* found in many natural networks, where a few nodes dominate with many connections. The construction of the BA model used in this work is as follows:

  1. Generate an ER random network with the number of nodes $N_0 = 10$ and connection probability $q = 0.5$ as the initial network of the BA model.
  2. At every time step, a single new node is introduced to the network.
  3. The new node connects to $n$ existing nodes in the network. The existing nodes are selected by the rule of preferential attachment, in which the nodes are selected with the probability proportional to their degree. Mathematically, the probability $P_a$ that a new node added at time step $t$ connects to an existing node $a$ is given by:

  $$P_a = \frac{k_a}{\sum_{b=1}^{N_t} k_b},$$

  where $k_a$ is the degree of node $a$, $N_t$ is the number of the existing nodes at time step $t$, and $k_b$ is the degree of node $b$.
  4. Iterate steps 2 and 3 until all nodes and edges are added.

- **Popularity-similarity-optimization (PSO) model**: The PSO model is designed to capture both popularity and similarity between nodes. Nodes are embedded in hyperbolic space, where each new node introduced at time step $t$ is assigned a radial coordinate $r = \ln t$ and a random angular coordinate. The connection between the new node and existing nodes is determined by minimizing the product of the birth time $s$ of the existing nodes and the angular distance $\theta_{st}$ between the new and existing nodes, such that nodes with the smallest $s\theta_{st}$ values are selected to connect. The radial distance corresponds to node popularity, while the angular distance represents their similarity. The parameters used in this work are set as $m = 5$, $L = 5$, $\gamma = 2.1$, $T = 0.4$, and $\zeta = 1$. This model effectively simulates social networks where both popularity and shared interests influence connections.

- **Fitness model**: In the Fitness model, each node is characterized by a *fitness value* $\eta_i$, reflecting its intrinsic ability to attract connections. The probability of a new node connecting to an existing node is given by:

  $$P_a = \frac{\eta_a k_a}{\sum_{b=1}^{N_t} \eta_b k_b}$$

  where $\eta_a$ and $\eta_b$ represent the fitness of nodes $a$ and $b$, respectively, and $k_a$ is the degree of node $a$ at time $t$. The fitness values are drawn from a power-law distribution, $P(\eta) = \eta^{-3.5}$, emphasizing the heterogeneity in node attractiveness. This model is commonly used to simulate networks where both competitiveness and popularity influence connections, such as academic citation or business networks.

