# OpenReview forum: "A Diffusive Data Augmentation Framework for Reconstruction of Complex Network Evolutionary History"
_ICLR.cc/2025/Conference — ICLR 2025 Conference Withdrawn Submission_

### Official Review · Reviewer_nheS · 2024-10-24

**Soundness:** 2
**Presentation:** 1
**Contribution:** 2
**Rating:** 3
**Confidence:** 4

**Summary:**

The paper proposes a new training framework for a model that predicts the appearance between two edges. The model itself corresponds to a vanilla neural network of 3 layers. The input layer receives the embeddings of two edges, the middle layer uses ReLU as the activation function, and the output is just two neurons with a scalar which are converted to a probability using a softmax function. Originally, one network is used to train the model. From this network, 100 timestamps are generated and used in the training process. The paper proposes the use of a diffusion model to replicate the source network and be considered in the process (instead/in conjunction with the source network). The results are promising as they show that the model is also able to replicate the generation process of a network.

**Strengths:**

The introduction of the paper shows a very important problem, try to infer the timestamps of the edges to learn the generate mechanism of a network. The preliminaries are well-explained and have all the details to understand the basic part of the paper. Subsection 3.1 defines the main model (the vanilla neural network), explaining the technical details based on the dimensions of the embedding. Unfortunately, the other subsections and sections must be improved in terms of clarity.

The explanation of the baselines in the appendix is a very good approach to save space, congrats.

The significance of the paper could be very high if all the processes are correctly applied. However, because of the lack of information, it is impossible to assess a more detailed evaluation of the strengths of the paper.

**Weaknesses:**

The originality of the paper is low. The use of a diffusion model to enhance a training process has been applied to several other problems. Maybe, the original part is given by the application of the diffusion model to infer the timestamp of edges. Unfortunately, this is not the real problem.

The clarity of the paper must be largely improved. While the first section allows the understanding of several details, other details are omitted, making the paper difficult to understand (for example, the problem itself). The conclusion mentions "Our work focuses on predicting edge generation times from given network structures, which aids in understanding network evolution and forecasting future developmental trends"; however, the proposed model (vanilla neural network) determines the probability of an edge over another, not the edge generation time. This implies, that the real problem is the selection of an edge over another given the current state of the network. This lack of information is also observed in the training and evaluation process, making the paper impossible to replicate.

The paper is not self-contained, the experiment section is barely described, and most of the work is given to the reader through the phrase "In this experiment, conducted under the CPNN framework, we generated augmented temporal networks using both the TIGGER method and our diffusion-based approach". Unfortunately, this is not enough to understand the training process and the evaluation process. I understand the generation of 100 timestamps from a single network, and that they are used for the training process. However, the generation data for the training process is not explained in detail. Do you generate the output based on two different timestamp networks? Do you generate |E_i|-|E_{I-1}| data points for the training process per timestamp? Similarly, the generation network process applied in the experiments section is not explained. Do you start with an empty network, or with 50% of the edges (like the sampling strategy described in subsection 3.3)? Given a network, how do you generate the next edge? Do you try all possible edges and compare among all of them, or do you use another strategy?

The paper mentions "improves the prediction accuracy for edge generation order across various networks". However, the experiment is not explained. Do you use an entire network and take two different edges? Are they consecutive edges or one of them is in the network and the other one is not? Do you use a similar approach to the timestamp generation and consider only the edges that were generated?

Section 3.4 explains the graph diffusion model. Unfortunately, this section seems a very general description of the difussion model instead of a description focused on the main contribution of this work. The main relation with the proposal is the last part of the subsection "First, a pure Gaussian noise is sampled and then the denoising networks iteratively predict the noise to be removed, and the ordered edges will be obtained from the weights of the sampled network gradually".

As can be observed the replicability of the paper is very low, and all these details reduce the quality and clarity of the paper.

Figure 2 must be changed. The final output says "Edge time prediction", but it determines the output between two possible edges. This is repeated multiple times throughout the paper. For example, it says "generation time prediction accuracy". However, the final model receives the embeddings of two edges, and, after the application of a softmax function, determines the probability of which edge should be added to the network. Also, why do you use two neurons with linear functions instead of using two neurons with softmax?

The details of the networks used in the experiments are not included in the paper (other than the number of nodes of the synthetic networks).

The paper uses a trainingNet and a TargetNet. This is barely described in the experiment section. Something is mentioned in subsection "3.2".1.

According to the results, the number of augmented networks does not increase the performance (subsection 4.4). Did you try the model without any augmented network (or this is equivalent to a baseline)?

Finally, the results from subsection 4.6 are alarming. According to the paper, the model can generate almost the same network. This implies a clear overfitting of the process. Unfortunately, this can not be evaluated because of the lack of details mentioned previously.

**Questions:**

Please see the multiple questions mentioned in the previous section.

---

### Official Review · Reviewer_3ihJ · 2024-11-02

**Soundness:** 3
**Presentation:** 3
**Contribution:** 2
**Rating:** 6
**Confidence:** 2

**Summary:**

The authors present an interesting paper on the creation of augmented networks to train a model to recreate the history of networks using a denoising diffusion model. The paper is well written, and the task seems to be of some interest. However, I am not a real expert in the domain, so my review should be taken with a grain of salt.

**Strengths:**

The presentation is clear (even a reviewer not expert in network history reconstruction could understand clearly what was done).
The idea of creating a large number of examples of a network for reproducing its history seems logical
The method seems to be as good as state of the art and perhaps better

**Weaknesses:**

The results are not presented with any statistical test
It is not clear what the accuracy measure represents (accuracy of edge generation order), nor how it is measured in real world-networks
The details of the sampling are not very clear.

**Questions:**

I would appreciate a better explanation of the success metric?
Is the improvement statistically significant?
In a real life situation, what would be the advantage of creating such an order (which is even in the best case still often wrong)?

---

### Official Review · Reviewer_dyYV · 2024-11-04

**Soundness:** 3
**Presentation:** 3
**Contribution:** 2
**Rating:** 5
**Confidence:** 4

**Summary:**

The authors propose a framework to enhance the reconstruction of evolutionary history in complex networks, focusing on accurately predicting edge generation times. The authors introduce a novel approach using diffusion models, specifically a model termed TopoEvoDiff, which generates temporal networks to augment scarce datasets. Their method aims to mitigate issues such as overfitting and limited generalization, which are common in network evolution studies due to the scarcity of temporally labeled data.

**Strengths:**

- The paper effectively employs diffusion models for data augmentation, addressing a significant challenge in network evolution reconstruction—namely, the limited availability of temporally labeled network data.

- The proposed model demonstrated a notable improvement in prediction accuracy highlighting its potential efficacy across diverse types of complex networks, from biological networks to social and collaboration networks.

- The framework eliminates the need for retraining on specific network types, suggesting substantial efficiency in both time and resource use, as evidenced by its ability to generalize across different network types and sizes.

**Weaknesses:**

- Limited real world experiments, Including more varied, real-world datasets could demonstrate the model’s adaptability to different domains, enhancing the paper’s appeal and impact. Specifically, the paper uses only 4 real world network making the experimental set-up not so convincing.

- Lack of comparison with other generative models, a broader comparison with other data augmentation or graph generation techniques, such as GANs, could offer insights into the unique advantages of diffusion models over alternative methods in this context.

**Questions:**

Could the authors provide us with more real world experiments of more diverse and additional networks, such as a bigger collection of social networks, PPI networks, rather than just 4 datasets.

---

### Official Review · Reviewer_3Hz3 · 2024-11-04

**Soundness:** 2
**Presentation:** 2
**Contribution:** 2
**Rating:** 1
**Confidence:** 4

**Summary:**

The authors propose an approach to predict the order in which edges were formed in a graph without any time information. This is an important problem in complex networks, as the times at which edges were formed often provides a lot of insight into the behavior of the networked system, possibly as much as the edges themselves. The authors formulate the edge time prediction task as a pairwise ordering prediction between each pair of edges. The main contribution appears to be a data augmentation approach using edge sampling and diffusion models to generate new temporal graphs. The authors present mixed results on synthetic and real networks.

**Strengths:**

- Considers an important problem in network science that has not yet received significant attention from the graph learning community.
- Proposed framework is relatively easy to understand.
- Moderately novel, as it uses standard diffusion models for graphs, but for a new task.

**Weaknesses:**

- The authors characterize their approach as edge time prediction. But they don't actually predict any edge times--only the order of edge formation. This is not the same task at all! For example, consider the difference between Peixoto & Rosvall (2017), who model temporal networks as a sequence, compared to Junuthula et al. (2019), who model the actual edge formation times using a temporal point process model. Peixoto & Rosvall (2017) provide a potential path to link the two approaches by modeling waiting time distributions in addition to the sequence--the authors may wish to consider a similar approach.
- The presentation is quite sloppy and lacks detail in many key areas. For example, they don't even describe how they perform the edge sampling--see question 1 below.
- Claims are not supported by evidence. See question 3 below for one example.
- Proposed evaluation metric is not interpretable. See question 4 below.

Sampling of presentation issues:
- Lines 238-239: proofreading and revising required: "where introduce the meaning of symbols, t: diffusion steps, N: normal distribution, betat: noise
level at step t, I: identity matrix."
- Lines 257-258: wrong dot used inside the L2 norm.
- Caption for Table 1 does not explain what the quantity in parentheses is. I assume it is relative improvement compared to CPNN. Furthermore, the table is shrunken soo small that many of the quantities are barely readable. I suggest moving some of the results to the supplementary material to allow a larger font size in the table.

References:
- Junuthula, R. R., Haghdan, M., Xu, K. S., & Devabhaktuni, V. K. (2019). The Block Point Process Model for continuous-time event-based dynamic networks. In Proceedings of the World Wide Web Conference (pp. 829-839).
- Peixoto, T. P., & Rosvall, M. (2017). Modelling sequences and temporal networks with dynamic community structures. Nature Communications, 8(1), 582. doi:10.1038/s41467-017-00148-9

**Questions:**

1. How is the sampling of edges done? Just randomly selecting 50% of the edges? Some type of random walk starting at a node?
2. Why is relative improvement compared to CPNN a useful quantity to assess performance?
3. Figure 4 suggests that the diffusion model is generating networks with extremely high clustering coefficients. The clustering coefficients of the original networks are not visible in the figure so there is nothing to compare to, so how can we verify your statement that "clustering coefficient distributions closely match
between the generated and original networks, reflecting consistent local clustering tendencies"? Furthermore, extremely high clustering coefficients in the range of 0.8-1.0 are almost never seen in real networks. I would argue that the lower clustering coefficients generated by the AAAI model are more representative of real networks.
4. What is a good value to obtain for your relative ranking metric shown in Table 1?

---

### Note · Authors · 2024-12-04

**Comment:**

Dear Program Committee Members,

I hope this message finds you well. I am writing to formally request the withdrawal of our submitted manuscript.

Over the past 2 weeks, it has become evident that several critical experiments and analyses are still required to comprehensively validate and enhance the findings presented in our paper. Despite our best efforts during the past two weeks, we have not been able to complete these experiments to the standard we aspire to. We strongly believe that the additional work will significantly improve the quality and impact of our research, ensuring it is more robust and complete for future dissemination.

We deeply appreciate the opportunity to submit to ICLR and sincerely apologize for any inconvenience this withdrawal may cause. Thank you for your understanding and for the time and effort the reviewers and the committee have invested in our work.

We look forward to resubmitting a thoroughly revised and improved version in the future.

Thank you for your kind consideration.

Best regards,
En XU

**Withdrawal Confirmation:**

I have read and agree with the venue's withdrawal policy on behalf of myself and my co-authors.